# Comprehensive Transcriptome Analysis Reveals Genome-Wide Changes Associated with Endoplasmic Reticulum (ER) Stress in Potato (*Solanum tuberosum* L.)

**DOI:** 10.3390/ijms232213795

**Published:** 2022-11-09

**Authors:** Venura Herath, Jeanmarie Verchot

**Affiliations:** 1Department of Agriculture Biology, Faculty of Agriculture, University of Peradeniya, Peradeniya 20400, Sri Lanka; 2Department of Plant Pathology and Microbiology, Texas A&M University, College Station, TX 77802, USA

**Keywords:** RNA-seq, *Solanum tuberosum*, endoplasmic reticulum (ER) stress, unfolded protein response (UPR), potato genome

## Abstract

We treated potato (*Solanum tuberosum* L.) plantlets with TM and performed gene expression studies to identify genome-wide changes associated with endoplasmic reticulum (ER) stress and the unfolded protein response (UPR). An extensive network of responses was identified, including chromatin remodeling, transcriptional reprogramming, as well as changes in the structural components of the endomembrane network system. Limited genome-wide changes in alternative RNA splicing patterns of protein-coding transcripts were also discovered. Significant changes in RNA metabolism, components of the translation machinery, as well as factors involved in protein folding and maturation occurred, which included a broader set of genes than expected based on Arabidopsis research. Antioxidant defenses and oxygen metabolic enzymes are differentially regulated, which is expected of cells that may be experiencing oxidative stress or adapting to protect proteins from oxidation. Surges in protein kinase expression indicated early signal transduction events. This study shows early genomic responses including an array of differentially expressed genes that have not been reported in Arabidopsis. These data describe novel ER stress responses in a solanaceous host.

## 1. Introduction

Environmentally introduced cellular stress or developmental programming can impact the endoplasmic reticulum (ER) and disturb the protein synthesis and maturation machinery including influencing translation efficiency and disrupting co- and post- translational protein folding [1,2,3,4]. The unfolded protein response (UPR) manages the accumulation of malformed proteins by bolstering the successful maturation and secretion of cellular proteins toward maintaining cellular homeostasis [5]. In Arabidopsis, the UPR is associated with major cellular activities involved in pollen development, root growth, abiotic stress responses, and responses to pathogen invasion including innate immunity [6].

The inositol-requiring enzyme (IRE1) is the most conserved transmembrane sensor found in mammals, yeast, and plants and is normally bound in the ER lumen by the binding immunoglobulin protein (BiP) [1,2]. Dissociation of BiP leads to IRE1 oligomerization and autophosphorylation. The cytosolic face of IRE1 has endonuclease activity that unconventionally splices the mRNA for the bZIP60 transcription factor in plants and XBP1 in mammals [3]. The active bZIP60/XBP1 factors translocate to the nucleus and drive the expression of ER-resident chaperones and other UPR-associated genes. IRE1 also engages in bulk degradation of mRNAs through a process known as regulated IRE1-dependent decay (RIDD) [4,5,6,7]. Alongside the IRE1/bZIP60 pathway in plants are the bZIP17/bZIP28-led pathways [4,8,9]. In Arabidopsis, AtbZIP17 and AtbZIP28 are transmembrane sensors that normally reside along the ER. Upon their activation, these factors move to the Golgi, where the S1P and S2P proteases remove their transmembrane domains. The active AtbZIP17 and AtbZIP28 transcription factors also drive expression of ER-resident chaperones and other UPR-associated genes [9,10,11,12]. AtbZIP60, AtbZIP17, and AtbZIP28 can heterodimerize broadening their number of gene targets [9,13,14,15]. 

Pharmacological agents that can induce ER stress and the UPR pathway have been widely used in mammals, yeast, and plants to probe the UPR pathways to understand the stress signaling mechanisms. Application of tunicamycin (TM) to cells and tissues has been valuable for advancing our knowledge of (a) the ER stress sensors and transcriptional changes that ensue, (b) factors involved in ER-to-nuclear signaling, (c) the degradation of mRNAs that localize to the ER during the UPR, (d) ER-associated cell death regulation, and (e) autophagy [8,11,16,17,18,19,20,21]. TM treatment of mammalian, plant, and yeast cells or tissues has been instrumental in demonstrating the evolutionary conservation of the proximal ER stress sensors that regulate separate but intertwined signaling cascades. TM treatment was employed to demonstrate that IRE1 autophosphorylation regulates its endonuclease activity [22]. For studying TM sensitivity and early genetic responses in roots, researchers have often treated Arabidopsis seedlings with TM and transferred them to fresh media to monitor root growth and stress recovery [11,23,24]. For chronic ER stress studies, researchers have also directly grown Arabidopsis seedlings on plates containing TM, and then harvested samples over a longer period to understand genetic responses contributing to TM resistance [19,25]. Arabidopsis has more than 250 ER stress-responsive genes detected within 2 to 10 h following treatment with TM, and many are upregulated [26,27]. These data represent UPR responses that restore cellular homeostasis after temporary stress. Transcriptomic studies performed in yeast and Arabidopsis revealed that most of the UPR target genes encode for: a) ER-resident chaperones, b) components of the secretory pathway, and c) components of the ER-associated degradation (ERAD) machinery [28] 

TM treatment was also valuable for RNA-seq studies in soybean and the characterization of stress-responsive GmNAC transcription factors [29]. Transcriptome analysis of TM-treated maize seedlings provided insight into the temporal changes in plant gene expression involved in transitioning cells from survival to cell death [19]. TM treatment (2 hr) of rice seedlings identified 374 ER stress-responsive transcripts including novel ER stress-responsive genes [30]. Notably, the soybean, maize, and potato orthologs for *IRE1*, *bZIP17,* and *bZIP28* have similar names [17,31]. The genes encoding the membrane-associated bZIP transcription factors in rice, OSbZIP39 and OsbZIP60, are orthologous to *AtbZIP17* and *AtbZIP28*.

Potato ranks in the top four most important crops in the world and grows in all environments and hemispheres of the world. There is a need to understand the landscape of gene families that influence growth, development, and adaptive stress responses. The current available complete genome sequence, genome annotation, and transcriptomes [32,33,34,35,36] for the *S. tuberosum* Group Phureja, *S. stenotomum*, and several cultivated varieties allowed us to perform such comprehensive identification and analyses and to conduct comparisons with Arabidopsis gene families. We identified three *BiP* homologs in the potato genome using phylogenetic, amino acid sequence, 3-D protein modeling, and gene structure analyses involving comparisons with the Arabidopsis *BiP* genes. *Ab initio* promoter analysis revealed the key role of promoter architecture in *BiP* gene diversity [37]. The recent characterization of the *StbZIP* transcription factor gene family also provided functional and regulatory classification groups based on the framework for the classification of *AtbZIP* family members [38,39,40]. Arabidopsis has seventy-eight *bZIP* family member genes and potato has eighty *bZIP* genes. The *bZIP* functional groups were identified as A through N, plus S [41]. Groups B and K include ER stress-responsive *bZIPs* and for Arabidopsis, these are *AtbZIP17, AtbZIP28, AtbZIP49,* and *AtbZIP60*. In potato, these are *StbZIP17, StbZIP28*, *StbZIP33, StbZIP60, StbZIP67, StbZIP70* and *StbZIP71*. The expanded numbers of ER stress-responsive *StbZIP* genes relative to *AtbZIPs* in the same clades suggest that stress adaptation in potato plants includes unique factors that may not be present in Arabidopsis plants. 

The goal of this study is to identify ER stress-responsive genes and the components of the UPR gene network in potato, *S. tuberosum*. Foliar TM sensitivity experiments were performed by treating leaves with TM and then harvesting samples within hours of treatment to study immediate changes in gene expression related to ER stress recovery by qRT-PCR [42,43,44]. In this study, we treated potato leaves with TM and performed gene expression studies to identify differentially expressed genes (DEGs) involved in TM sensitivity and ER stress recovery.

## 2. Results

### 2.1. Expression Profiles of Genes Involved in the Early ER Stress Response

We obtained detailed information about the expression profiles of genes involved in the initial stages of ER stress response in potato leaves (cv. Russet Norkotah) by comparing the expression profiles of plants treated with TM versus treatment with DMSO (solvent only) for 2 and 5 h. Using BGI RNA-seq technology, between 37 and 38 million quality read pairs aligned with the Castle Russet potato genome [32], and the overall alignment rate was greater than 94% (Appendix A). Between 21.3 and 22.8 million read pairs aligned uniquely with the genome representing 56 to 62% of the total reads. Between 14.1 million (39 %) and 15.8 million (43%) read pairs returned multiple hits. Between 703,240 (1.9%) and 886,837 (2.4%) read pairs did not align to the reference genome.

Volcano plots visually represent the differential gene expression analysis (Figure 1A). TM treatment for 2 and 5 h produced a total of 806 unique genes that were differentially expressed (Figure 1A) with a threshold fold change of 1.0 and *p* < 0.05. There were 204 uniquely upregulated and 278 uniquely downregulated genes at 2 h, 157 uniquely upregulated, and 129 uniquely downregulated genes at 5 h. Among the DEGs, nine genes were upregulated at 2 and 5 h, nine were downregulated at 2 h but upregulated at 5 h, nine were upregulated at 2 h but downregulated at 5 h, and ten were downregulated at 2 and 5 h (Figure 1B). The Castle Russet locus IDs and the putative orthologs found in the double monoploid Phureja DM1-3 genome [32,33] representing each of the DEGs are provided in Appendix A with the log_2_ fold change, adjusted *p*-values, false discovery rate (FDR), and annotations including the predicted common gene names. Appendix A presents genes that are upregulated at 2 h, 5 h, and both 2 and 5 h. Appendix A presents genes that are downregulated at 2 h, 5 h, and both 2 and 5 h. Appendix A provides oppositely regulated genes at 2 and 5 h. 

To best understand the relationship of these DEGs to biological processes, molecular functions, and cell components, we used the BLAST2GO tool built into OmicsBox to chart the overall distribution of gene ontology (GO) terms for three categories of GO level 2: Biological Process, Molecular Function, and Cellular Component. Regarding biological processes, the highest number of GOs/Seq length at 2 and 5 h contributed to primary metabolic processes, organic substance metabolism, nitrogen metabolism, cellular metabolism, and biosynthetic processes (Figure 2A,B). Intermediate annotation score distributions were attributed to several related processes involving responses to external and endogenous stimuli, including biotic and abiotic stimuli, signal transduction, and cell communication, as expected for tissues undergoing ER stress. In the category of molecular functions, the highest GOs/Seq length at 2 and 5 h included heterocyclic and organic cyclic compound binding (i.e. noncovalent binding), as well as small molecule binding. The intermediate score distributions for molecular functions were transferases, hydrolases, and protein-binding, which points to an increasing need for stress-responsive and protein-folding enzymes. The lowest number of GOs/Seq length at 2 and 5 h represented lipid/carbohydrate binding, transcription factors, translation regulators, signaling receptors, and chromatin binding factors (Figure 2A,B). Cellular components at 2 and 5 h include intracellular anatomical structures, organelle, cytoplasm, and membranes. As expected for tissues experiencing ER stress, the common GO terms associated with cellular membranes include endomembrane system, cell periphery, envelope, and membrane-enclosed lumen (Figure 2A,B).

Blast2GO functional enrichment analysis [45] was employed using Fisher’s exact test to assess the significance of the associations of the cellular components between the TM and mock-treated samples. The annotation distribution chart shows the enriched factors at 2 h associated with protein binding, precursor metabolic, Golgi apparatus, and cell cycle. At 5 h, the enriched factors associated with the endomembrane system, especially the ER and Golgi (Figure 3A). The ER-associated factors were approximately 2% to 7% of the enriched sequences. The Golgi-associated factors were elevated to 4% following TM treatment at 5 h relative to the mock dataset. Enrichment for the endomembrane and secretory system has also been reported in Arabidopsis treated with TM [26,27]. The GO enrichment analysis results at 2 and 5 h were also visualized using Word Cloud summaries. The sizes of the GO categories reflect the strength of the enrichment relative to other results in the query (Figure 3C,D). The major genes at 2 h were associated with cellular anatomy, cytoplasm, membranes, and the nucleus. At 5 h, transcripts associated with the endomembrane system, Golgi, ER, peroxisomes, organelles, and ribosomes were more abundant (Figure 3C,D). 

A major role of the UPR machinery is to manage the influx of newly synthesized proteins into the already stressed ER compartment [2,17,25,28], and to increase cellular secretory activities. To ascertain whether TM similarly influences nascent protein processing and cellular secretory activities in potato cells, we categorized DEGS (Appendix A) for their roles in the endomembrane network based on their GO terms, common names, and descriptions of gene function. Appendix A shows the appearance of at least 39 transcripts contributing to the structure and function of the ER, Golgi, endocytic, and vacuolar networks. These are factors primarily membrane embedded proteins. Notably at 2 h, the signal recognition particle (SRP) receptor beta subunit (Soltu.Cru.11_0G007340.6), which is instrumental in moving nascent peptides into the ER lumen [1,2,3], is upregulated (Appendix A). DEGS contributing to vesicle trafficking include ARF, COP adaptors, nucleoporins, and SEC family member proteins (Appendix A). Other prominent factors are cytochrome P450 family members; enzymes involved in peptide catabolic processes or proteolysis; and two genes involved in bacterial defense responses.

Chronic ER stress responses in eukaryotic cells are accompanied by changes in mitochondrial or peroxisomal oxidative metabolism, as well as autophagy [17,46,47,48,49,50,51]. In plants, peroxisomes play a significant role in oxidative metabolism and reactive oxygen species detoxification. In yeast and Arabidopsis cells, autophagosome formation accompanies UPR-inducing conditions [22,47,48,49,50,51]. Autophagosomes serve to move damaged proteins to the vacuole for degradation. Furthermore, the vacuole contributes to both autophagic degradation of proteins, and endocytic functions. In this informational context, we examined the transcriptome dataset to determine if TM treatment of potato leaf cells stimulates the expression of genes associated with oxidative stress, cell death, autophagy, and vacuolar trafficking. While there is little evidence for mitochondrial cell death pathways among the DEGs, we identified three autophagy-related factors among the downregulated genes at 5 h (Appendix A). Ten factors in Appendix A are primarily involved in trafficking to the vacuole or peroxisomes. It is possible that the regulation of cell death programming and autophagy is a later response to TM-induced ER stress and a longer time course is necessary to detect its’ activation.

Only two nuclear pore factors are present in the enriched dataset suggesting that this avenue of protein or mRNA trafficking is not as responsive to TM treatment as the intracellular protein transport machinery involving the secretory and endocytic network. Since environmental assaults trigger ER stress responses, it is notable that there were at least twenty-three factors associating with the plasma membrane or cell wall that showed altered expression, including factors that are linked to pathogen defense or cell wall modifications (Appendix A).

### 2.2. Chromatin Modifications and Altered Gene Regulation Are Early Stress Responses to TM Treatment

To obtain evidence that TM treatment alters gene activation or gene silencing, we examined the transcriptome dataset for chromatin remodeling factors, DNA-or histone-modifying enzymes that influence nucleosome stability and chromatin accessibility [52,53,54,55,56,57,58,59] (Appendix A). Among the upregulated genes are an H2A variant, four SET domain protein methyltransferases that act on histones, components of the SWI/SNF chromatin remodeling complexes that are important for regulating transcription, and nucleosome assembly proteins (NAP). NAP1 normally counteracts the SWR1 complex for H2A variant substitutions and is simultaneously downregulated, supporting the hypothesis that chromatin remodeling is an important early response [55,56,57]. Components of the histone acetyltransferase complex, regulating genes whose expression depends upon H2A and H4 acetylation, are downregulated at 2 hr. A putative chromatin modification-related protein *MEAF6* and a putative *histone deacetylase 9* are also downregulated [43,57,58,59]. DEGs that contribute to gene silencing and activation of the RNA-induced silencing complex (RISC) include an *SGS3-like family DNA methylation 1* gene (Soltu.Cru.03_4G020410.1); putative ROS1, which is a plant 5-methylcytosine DNA glycosylase (Soltu.Cru.10_1G005460.1); a translin-like factor (Soltu.Cru.05_2G012280.1); and a NERD-like zinc finger C3H domain-containing protein (Soltu.Cru.02_4G0002180.1) (Appendix A) [57,58]. The downregulated genes at 5 h encode putative DNA methylation-like factors, histone-modifying enzymes, and DNA-directed RNA polymerase subunits pointing toward reprogramming gene expression, and alternative splicing of pre-mRNAs from multi-exon genes [59]. These data suggest that chromatin remodeling for the purpose of gene activation or repression is a component of the stress response [60]. Furthermore, few factors associate with chromatin functions in the chloroplast. 

Several DEGs were identified in our study including DNA polymerase type-B family, replication factor A protein family 1, nucleosome assembly protein (NAP) family, single strand DNA binding protein precursor, and inactive poly(ADP-ribose) polymerase SRO4-related family [46,60,61,62] are potentially involved in genotoxic stress responses (Appendix A). We also identified factors that can be recruited for transcription and its regulation (Appendix A). Most interesting is the early upregulation of *MED25* (Soltu.Cru.12_4G008930.3), which is a key transcriptional coregulator participating in jasmonic acid (JA) and abscisic acid (ABA) signaling and gene expression [63,64]. Other transcriptional regulators that contribute to hormone and stress responses include the potential ARF family, *ARR-like* family, *bZIP* family, *CAMTA-like* family, *MYB* family, and *WRKY* family members (Appendix A) [65,66,67] These data further support a model in which chromatin remodeling and changes in gene activation result from TM treatment. Transcription factors and regulators that belong to families known for their involvement in cell cycle regulation and development are differentially regulated.

### 2.3. Changes in the Accumulation of Transcript Isoforms and Genes Involved in RNA Metabolism Are Seen in TM-Treated Leaves

The regulation of RNA pol II and histone modifications are fundamental to adjusting gene expression and linked to shifts in the fraction of mRNA isoforms expressed from individual genes [68]. Transcriptional regulation involves histone-lysine-N-methyltransferase enzymes that aid in transcription start site selection (TSSs) and transcription termination at intragenic polyadenylation sites. Given that the factors influencing chromatin structure and transcription in Appendix A can potentially influence alternative TSSs, we investigated the DEGs-dataset to identify alternative gene isoforms in TM-treated versus mock-treated leaves (Appendix A). We considered the possibility of alternative splicing (AS) of mRNA alongside alternative TSSs.

We identified nine loci at 2 h and fifteen loci at 5 h following TM treatment, where RNA isoform usage changed (Appendix A, Figure 4 and Figure 5). The alternative gene isoforms at 2 and 5 h appear to result from intragenic TSSs or alternative termination sites. The first and most notable observation at 2 h is a shift in isoform usage for four loci encoding proteins with intrinsically disordered regions (IDRs), including one reported regulatory component of DCP5 of the mRNA decapping complex (Soltu.Cru.05_1G002100). The mRNA decapping complex is central to the assembly of processing bodies (P-bodies) which consist of mRNA-ribonucleoprotein granules. Notably, at 5 h there is also a shift in isoforms for RanBP2-type zinc finger protein (Soltu.Cru.05_1G009100), an RNA binding protein that may also associate with P-bodies. Assembly of mRNA into P-bodies is associated with translational arrest and mRNA decay. P-bodies contain components of nonsense-mediated decay (NMD), zinc finger binding proteins that bind RNA, and RNA helicases. There are preferred isoforms in Figure 4 and Figure 5 that are subject to NMD. While there is a shift in isoform expression, there is not an overall change in gene expression for most genes studied at 2 h (Figure 4).

It is also noteworthy that transcripts presenting shifts in isoform usage include factors contributing to gene regulation, protein expression, or protein turnover. Among these are Soltu.Cru.01_2G011350, a putative transcription factor; Soltu.Cru.08_0G003780, a potential RNA splicing factor; Soltu.Cru.06_1G021440, a linker histone H1; Soltu.Cru.11_1G000670, an isoleucine-tRNA ligase; Soutu.Cru.05_1G007930, a component of the chloroplast SRP insertion system; and Soltu.Cru.03_0G014040, defective in cullin neddylation protein (Appendix A). At 2 and 5 h, transporters and synthases were among the genes that also show altered RNA isoform usage.

Appendix A shows an enrichment of genes primarily involved in mRNA/rRNA/tRNA processing, ribosome biogenesis and assembly, and nuclear transport, indicating extensive regulation of RNA metabolism early in ER stress induced by Tm. To better highlight these gene functions Appendix A includes categories based on GO terms, common names that can be found by BLAST search, and UniProt descriptions. Few differentially expressed genes associate with the chloroplast or mitochondria while the majority appear to associate with the nucleolar, nucleoplasm, and cytoplasmic processes surrounding mRNA, rRNA and tRNA functions. At 2 and 5 h, at least 28 factors involved in mRNA synthetic processes, 21 factors involved in rRNA processing or ribosome biogenesis, and 32 factors involved in tRNA modifications and mRNA translational processes (Appendix A). These data support a model in which bulk protein synthesis is altered following TM treatment.

It is well established that the heterogeneous nature of multi-protein ribosome complexes is central to the regulation of mRNA translation [3,69,70,71,72]. Appendix A reveals significant changes to the 40S and 60S ribosome protein and ribosome associated protein paralogs that are available indicating that TM treatment stimulates changes in the regulation of mRNA translation, perhaps to acclimate the cellular proteome to stressful conditions [71,72]. These changes potentially influence the rate of translation or the nature of ribosome stalling on mRNAs which ultimately influences co- or post-translational folding of proteins, either in a positive or negative manner [2,73]. Further indication that the regulation of mRNA translation is altered by TM treatment include observations that factors involved in tRNA maturation, tRNA aminoacylation, translation initiation and translation elongation were also enriched (Appendix A). These data suggest that TM-induced UPR is coupled to changes in translation through the coordinated dynamic changes in the heterogeneity of ribosomes and tRNA processes.

### 2.4. Gene Expression Involving Protein Maturation and Degradation Pathways

Proteostasis is the regulation of protein folding and elimination of malformed proteins to ensure correct translation, maturation, and subcellular targeting of proteins which are critical for cellular metabolism [13]. TM treatment inhibits N-glycosylation of proteins and thereby disrupt proteostasis in the ER [27,71,72,73,74]. IRE1 dimerization is the first most crucial step to activation of UPR signaling, which is regulated by BiP, and offers reversible chaperone repression of IRE1 activity. BiP, an Hsp70 chaperone, has intrinsic ATPase activity accelerated by DNAJ-protein co-chaperones. In particular, the mammalian cell ERdj4 is shown to selectively repress IRE1 through its association with BiP [1,74]. *BiP3* (Soltu.Cru.01_3G025430.1) is upregulated at 5 h (Appendix A). IRE1 endonuclease activity functions to remove an unconventional intron in the *bZIP60* mRNA enabling the translation of the nuclear transcription factor, and to regulate the decay of mRNAs or miRNAs, via RIDD [3,4,5,6,7,75]. Regarding bZIP60-led gene expression, we expect to observe an increase in the expression of factors involved in protein folding and maturation [76]. Appendix A lists 106 genes that are differentially expressed and participate in nascent protein folding and maturation; protein modifications including N-linked glycosylation, (de)phosphorylation; ubiquitination and sumoylation; and proteolytic processes.

Among the enriched genes at 2 and 5 h are three genes encoding DNAJ-like chaperones that localize to the ER and plastids (Soltu.Cru.03_4G020150.1, Soltu.Cru.03_0G016890.1, Soltu.Cru.03_4G020150.1). Other UPR-associated chaperones that were identified include *calreticulin* (CRT; Soltu.Cru.05_2G013530.1) and its co-chaperone *Pollen Defective in Guidance 1* (Soltu.Cru.06_4G006080.1) [77]. Enzymes involved in nascent protein folding (Soltu.Cru.10_4G008250.1) and glycosyl modifications to proteins in the ER and Golgi (Soltu.Cru.06_3G011660.1 and Soltu.Cru.08_0G016470.1) were enriched in the dataset. Another group of ER-specific molecular chaperones is the peptidyl-prolyl cis-trans isomerases (PPIases) which catalyze the isomerization of prolyl bonds [2,78]. PPIases are important components of the protein folding machinery and there is a single *cyclophilin-type PPIase* (Soltu.Cru.06_1G019900.1) and an *FKBP-type PPIase* (Soltu.Cru.10_2G012040.3) that also identified. There are also eight protein chaperones and co-chaperones needed for enhanced protein folding capacity in the chloroplast and mitochondria (Appendix A) [73,76,79,80].

The ubiquitin-proteasome machinery and vacuolar proteases are responsible for eliminating misfolded or damaged proteins. We identified intramembrane enzymes that facilitate proteolytic maturation of membrane-embedded proteins. These enzymes belong to families of aspartic acid proteases, serine proteases, and metalloproteases. We identified at least twenty-six genes in the dataset that encode ubiquitin-conjugating enzymes, polyubiquitin, and proteasome subunits, suggesting the elimination of malformed proteins through the ubiquitin-proteasome system is one of the earliest responses to TM-induced ER stress. Nuclear enzymes associated with sumoylation are among the enriched factors in Appendix A. We identified 32 DEGs encoding proteases, peptidases, and inhibitors associating with the ER or other organelles (Appendix A) [77,81,82]. Notably, the upregulated protease inhibitors serve to prevent unwanted proteolysis and contribute to cellular defenses. Many Kunitz-type protease inhibitors are known for having bactericidal or insecticidal activities [78,82,83] and, we identified 14 DEGs that encode 14 protease inhibitors in Appendix A. 

### 2.5. Changes in Oxygen Metabolic Enzymes and Protein Kinases

We investigated the DEGs encoding oxygen metabolic enzymes and antioxidant defenses in Appendix A. This investigation was predicated on the DEGs involved in oxygen metabolism and regulation in Appendix A, changes in the RNA isoform usage for a gene encoding a nucleoredoxin (Soltu.Cru.05_2G006290) in Appendix A, and the peptide methionine sulfoxide reductase (Soltu.Cru.02_0G016970.2) in Appendix A. We observed a conspicuous shift in the expression of 14 genes encoding NADP-associated oxidoreductase enzymes occurring primarily in the mitochondria and chloroplast. Approximately 10 *peroxidases* involved in hydrogen peroxide removal were downregulated at 2 h but not at 5 h. In addition, cytochrome P450 pathway enzymes, which contribute to various metabolic pathways, are enriched at 2 and 5 h. Three cytochrome P450 members that are differentially expressed at 2 h belong to the CYP86 clan, which associate with the hydroxylation of fatty acids [81].

By searching the integrated annotations for the Eukaryotic protein Kinase and protein Phosphatase Database (iEKPD) [77] we identified 32 genes encoding protein kinases (Appendix A) that are differentially regulated at 2 and 5 h indicating significant signal transduction activity is stimulated by TM treatment. Fifteen kinases are members of the Tyrosine Kinase or Tyrosine Kinase-like families. Seven kinases are CMGC family representing cyclin-dependent kinases, and four are STE7 or STE11 family kinases, which contribute to the mitogen-activated kinase signaling pathways. To better understand their biological functions, we searched the GO terms for biological processes and cellular components in the SpudDB. The majority are described as responsive to abiotic stress and/or associated with cellular protein modifications. These kinases occur in the plasma membrane, cytoplasm, and nucleus, with three identified in the chloroplast and one appearing in the mitochondria. These combined data suggest that oxidative stress and signal transduction events are among the earliest transcriptional responses to TM treatment in potato leaves.

### 2.6. DEGs That Are Common between Potato and Arabidopsis Treated with TM

Prior transcriptomic studies analyzed the DEGs in TM-treated Arabidopsis plants and identified 259 TM-responsive genes [27]. We performed reciprocal BLAST to compare the DEGs at 2 and 5 h from potato leaves with the DEGs from Arabidopsis seedlings treated with TM for 2 and 5 h, by pooling the Arabidopsis DEGs, which were provided by Iwata et al. (2010) and presented the data using an UpSet plot (Figure 6). As a shorthand for representing the comparisons of *S. tuberosum* and *A. thaliana* datasets, we used S.t. x A. t. indicating DEGs that are upregulated (Up x Up), downregulated (Down x Down), or differentially regulated (Up x Down, Down x Up) in the datasets. Overall, 259 genes were upregulated in Arabidopsis, and 193 were homologous to upregulated genes in potato leaves. Approximately 150 Arabidopsis genes were downregulated, and 104 potato genes were downregulated. Twelve genes at 2 h and 5 genes at 5 h were upregulated in potato and Arabidopsis. Fifteen potato genes at 2 h and four potato genes at 5 h had homologs in Arabidopsis that were also downregulated. Surprisingly, 16 potato genes at 2 h and 3 potato genes at 5 h were downregulated, but their Arabidopsis homologs were upregulated. The total of 38 potato genes residing in multiple categories of up- and downregulated genes can be explained by the expansion of the potato genome and gene families relative to the Arabidopsis genome and gene families. For example, the Arabidopsis gene AT3G12900.1 is a member of the 2-oxoglutarate and Fe (II)-dependent oxygenase superfamily. Using reciprocal BLAST, we identify potential potato homologs which are differentially regulated at 2 and 5 h and account for four dots in the upset plot: Soltu.Cru.01_3G030430.1 is upregulated at 2 h, Soltu.Cru.12_1G018210.1 is downregulated at 2 h, Soltu.Cru.09_3G011280.1 is upregulated at 5 h, and Soltu.Cru.02_2G015520.1 is downregulated at 5 h. Thus, the upset plot features the simple common and unique genetic responses to TM treatment between potato and Arabidopsis leaves, as well as differences resulting from gene family expansion across evolution. 

## 3. Discussion

TM is used to investigate the UPR in eukaryotic models because it disrupts protein maturation in the ER via the inhibition of N-linked glycosylation [4,5,21,73,84]. Treatment of Arabidopsis seedlings with 2 or 5 µg/mL TM for 1–2 h is sufficient to observe activation of *AtbZIP60*, expression of *AtNAC103* induced by AtbZIP60, and transport of AtbZIP28 from the ER to the nucleus [3,21,27,80,85]. Transcriptome studies of maize seedlings and grapevine roots treated with 5 µg/mL TM for 2 or 3 h and 48 h showed clusters of UPR-associated genes surging at various hours post-treatment [71,84]. Here, in this study, we treated potato leaves with 5 µg/mL TM and collected transcriptome data at 2 and 5 h post-treatment. The volcano plots in Figure 1 as well as Appendix A present evidence that transient bursts of gene expression occur within the first hours following TM treatment, and a limited set of genes are commonly upregulated at 2 and 5 h. Evidence for effective TM-induced ER stress is also provided in Figure 3, which features the enrichment of genes regulating protein binding, precursor metabolism, Golgi apparatus, and ER. Similar observations were reported in yeast, where the volume of the ER network expands significantly and is an important precursor to UPR signaling in parallel to the changes in the expression of ER-resident chaperones [22,51].

Although the identities of the genes differed significantly at 2 and 5 h, the GO distribution pointed to only a few gene categories produce surges in gene expression. The upregulation of various metabolic processes, as well as heterocyclic, organic cyclic, and small molecular binding factors suggest that the cellular adaptive program ensures metabolic homeostasis. We observed an expected increase in genes contributing to intracellular anatomical structures, organelles, and cytoplasm. Factors contributing to the endomembrane and secretory system were enriched as expected from previous reported outcomes of Arabidopsis treated with TM [21,78]. Among the DEGs in Appendix A are factors contributing to the proper functioning of the ER, Golgi, endosome, and vacuole and this might be expected if ER expansion or restructuring is a component response to TM-induced ER stress. In Arabidopsis, IRE1b-led signaling due to ER stress leads to activation of autophagy. The mammalian IRE1 activates XBP1 to induce apoptosis and autophagy [24,46,47,48,49,50,51]. We saw little evidence for activation of cell death or autophagy pathways in potato leaves between 2 and 5 h. This is despite evidence linking certain DEGs to oxidative stress metabolisms such as oxidoreductases, peroxidases, dismutases, and cytochrome P450 subunits (Appendix A). Elevated changes in the expression of these genes or evidence for chronic ER stress would be needed to observe changes in the regulation of cell death or autophagy-associated genes.

Surprisingly, *StbZIP17, StbZIP28*, and *StNAC089* were not among the DEGs in the first hours of TM treatment. This analysis of the TM-treated potato transcriptomic responses suggests that the immediate transcriptional changes does not require their upregulation. IRE1 and bZIP17/bZIP28 may become activated at the ER, leading to the upregulation of protein chaperones and foldases ahead of their transcriptional induction. Prior studies in Arabidopsis showed that the level of *bZIP60* and *bZIP17/bZIP28* induction can be 2-fold to achieve significant changes in downstream gene expression. One explanation is that the natural abundance of *StbZIP60, StbZIP17/StbZIP28* in potato leaves may be sufficient to enact significant transcriptional changes during the first hours of ER stress [86]. Alternative, the expanded number of individual members of the ER stress functioning group B and group K of the StbZIP transcription factor family [38] may redundantly contribute to ER stress regulation, thereby reducing the need for stimulating the expression of StbZIP60, StbZIP17, and StbZIP28 in the first few hours. 

Chromatin remodeling plays an important regulatory role in gene activation and gene silencing in plants when faced with environmental stresses [59]. Appendix A provides evidence of genetic and epigenetic reprogramming occurring in TM-treated potato leaves, including DEGs never featured in previous studies of TM-treated Arabidopsis. These data support a model for structural changes to nucleosomes and chromatin that set the stage for rapid and transient changes in gene expression or epigenetic responses following TM treatment [60]. Nucleosomes contain DNA wrapped around an octamer of core histone partners, H2A, H2B, H3 and H4 histones. The types of remodeling that can lead to changes in gene expression include removing/shifting histones or introducing histone variants [52,53]. Such changes in histone–DNA interactions involves ATP hydrolysis. Indeed, H3 and H4 comprise the core histones, and exchangeable H2A and H2B variants can influence the accessibility of DNA for transcription factors, polymerases, and other nuclear proteins that regulate gene expression. Whereas the default regional chromatin state serves to limit RNA polymerase access, changes in histone–DNA interactions create accessible regions of DNA for polymerases, transcription factors, and other nuclear proteins that regulate gene expression. Covalent modifications such as ubiquitination, deacetylation, or methylation can generate heritable although reversible changes in gene expression, known as gene silencing. Covalent histone modifications require specialized enzymes, which were among the DEGs reported in this study. The increase in import Beta (Appendix A), which enables the nuclear import of proteins and miRNA loading into RISC complexes is a DEG that is required for successful gene silencing. Appendix A presents additional DEGs contributing to gene silencing and activation of the RNA-induced silencing complex (RISC) such as *SGS3-like family DNA methylation 1* (Soltu.Cru.03_4G020410.1), *translin-like factor* (Soltu.Cru.05_2G012280.1), and a *NERD-like zinc finger C3H domain-containing protein* (Soltu.Cru.02_4G0002180.1) genes. 

Plants are known to couple environmental cues to the production of alternative mRNA isoforms [57,68]. Beyond gene activation, gene repression, and gene silencing, the changes in nucleosome occupancy, and histone modifications such as methylation of N-tails, DNA methylation, and chromatin adapter complexes, represented by DEGs in Appendix A also influence the appearance of alternative mRNA isoforms. Added to this list are the *nucleosome assembly protein (NAP) family proteins, EAF6 family of chromatin modifying proteins, SET domain-containing proteins (i.e., histone-lysine N-methyltransferase), histone deacetylase, MORF-related genes, MED25,* and *DNA methylation-like 1* [53,54,55,56,60,61,62,63,64]. We identified examples where alternative isoforms arise from the alternative transcription start sites (ATSS) or alternative mRNA splicing (AS) in Appendix A and Figure 4, and while there are changes in the accumulation dynamics among these gene isoforms, no overall change in gene expression occurs. We examined the exon domains that appear in genes with significant isoform changes. We noted the genes showing changes in the intrinsically disordered regions (IDRs). It is worth speculating that as part of the UPR, the expression of IDR-containing alternative isoforms may be an adaptive response toward retaining key protein functionality and is necessary for cell survival while avoiding the need for rigid protein secondary, tertiary, or even quaternary structures [87]. More research in this area is needed to understand the contributions of alternative mRNA isoforms as well as IDRs to plant ER stress responses.

Across the Appendix A, there is evidence for several layers of dynamic changes that can influence RNA metabolism and other pre-ribosomal processes and point to shifts in the rate of translation [3]. Among the DEGs in Appendix A is a gene encoding a DNA-directed RNA pol III subunit, which is responsible for 5S rRNA transcription; and another encoding a subunit of pol II, that is required for transcription of ribosome-associated protein genes. Appendix A presents shifts in the isoform usage of mRNAs encoding an RNA decapping enzyme, an RNA splicing factor, and a tRNA ligase. Appendix A points to changes in the nuclear pore complex that influences RNA export which can bolster or attenuate translation. Furthermore, changes in the rate of translation or processing of nascent proteins can be influenced by changes in the subpopulation of ribosome subunit proteins, the absence of specific ribosome protein subunits, or the exchange of paralogs. Such changes typically influence the rate of translation and nascent protein folding. There are also shifts in the expression of aminoacyl-tRNA synthases, a tRNA ligase, and the downregulation of a translation initiation factor. Combined, these data support a model in which couples TM-induced UPR to changes in translation through the coordinated dynamic changes in the compositional heterogeneity of ribosomes and tRNA processes.

The UPR response involves the activation of factors involved in protein quality control. As expected, Appendix A features a range of protein chaperones, protein folding and modifying enzymes, and proteases involved in protein turnover. In Arabidopsis, reports indicate that *BiP, CRT*, and *PDI* are among the upregulated genes at 2 and 5 h, in potato leaves we see the expression of broader numbers of protein folding enzymes stimulated by TM treatment. The expression of the ubiquitin-proteasome machinery as well as SUMO modifying enzymes supports protein quality control. The protein quality control machinery also protects against protein oxidation and Appendix A features antioxidant defenses, peroxidases, and oxygen metabolizing enzymes. There are also changes in signal transduction as evidenced by the shifts in protein kinases involved in responses to environmental stresses.

## 4. Materials and Methods

### 4.1. Plant Material and TM Treatment

In vitro-grown *S. tuberosum* cultivar Russet Norkota were rooted in Root Riot^®^ peat-based cubes and kept in a growth room with a 12 hr photoperiod at 20 °C for four weeks. Three leaves of each plant were sprayed with 5 g/mL TM dissolved in DMSO (treatment) or DMSO (mock). Leaves were harvested after 2 and 5 h of exposure and immediately frozen in liquid nitrogen. Frozen samples were stored at −80 ℃ for RNA extraction and transcriptomic study. Three frozen leaves from TM or mock-treated plants were combined and ground for RNA extraction. Experiments were repeated three times.

### 4.2. Transcriptomic Analysis

Total RNA was extracted from pooled leaf samples using the RNeasy Mini Kit (Qiagen Co., Hilden, Germany). The Epoch 2 Microplate Spectrophotometer (BioTek Instruments Inc., Winooski, VT, USA) was used to assess RNA purity. The A_260_/A_280_ ratios of samples ranged between 1.9 and 2.1. The Agilent 2100 bioanalyzer (Agilent Technologies, Palo Alto, CA, USA) was used to assess RNA integrity and all samples had an RNA integrity number (RIN number) >7.3.

The mRNA purification, fragmentation, cDNA synthesis, second-strand synthesis, adapter ligation, cDNA library purification, and transcriptomic sequencing were performed at the Beijing Genomics Institute (BGI, Shenzhen, China) using the BGISEQ-500 platform according to Herath and Verchot, 2021 [86]. BGI performed PE150 strand-specific library preparation as follows. First, the poly-A-containing mRNAs were purified using oligo(dT)-coupled magnetic beads. Then, mRNA fragmentation was carried out using divalent cations under elevated temperatures. The fragments were converted to the first-strand cDNA using reverse transcriptase and random primers. To generate double-stranded cDNA, second-strand cDNA synthesis was carried out using DNA polymerase I and incorporating dUTP (2′-deoxyuridine 5′-triphosphate) in place of dTTP (2′-deoxyguanosine 5′-triphosphate). The final cDNA library was generated by purifying and PCR enriching the product from the earlier step. Using a rolling-circle replication mechanism, single-stranded DNA circles containing DNA nanoballs were generated. Then, the DNA nanoballs were loaded into patterned nanoarrays. Paired-end reads of 150 bp were generated with the BGISEQ-500. The raw data with adapter sequences or low-quality sequences were filtered using SOAPnuke (v2.1.0) [88]. FASTQC was used to assess read qualities (version 0.11.9). The subsequent analysis returned clean reads.

Reference-guided mapping was carried out using the Castle Russet Genome Assembly (Ver.2.0) included under the Phased Genome Assemblies and Annotation of Tetraploid Potato in SpudDB (http://spuddb.uga.edu/, last accessed 15 March 2022) [32,33,34,35,36,89] using HISAT2 (v2.2.1) [90]. The SAM files were converted to BAM files and indexed using SAMtools (v1.10) [90,91,92]. Assembly Alignment quality was assessed using FASTQC (v. 0.11.9) Transcripts assembly and abundance were determined using StringTie (v2.1.0) [93] and using the annotations obtained from the Castle Russet Genome Assembly (Ver.2.0). Raw sequence counts were calculated using HT-Seq (v2.0.1) [94]. Differential expression analysis was carried out using edgeR (v.3.38.1) [95] in RStudio Desktop (v.2022.02.3+492) or server (RStudio, Boston, MA, USA) hosted in the Texas A&M University high performance computing portal running R (v. 4.2.1) framework. Differentially regulated genes with ≤−1 or ≥1 log2-fold difference with a false discovery rate (FDR) of ≤0.05 at each time point were selected for further analysis. Volcano plots were generated using EnhancedVolcano (v. 1.14.0) [96]. All the scripts used for this study are available at https://github.com/venuraherath/TM-Transcriptome-Potato, accessed on 9 August 2022.

### 4.3. Gene Annotation and Characterization

Amino acid sequences of the differentially expressed genes were mapped and annotated using OmicsBox (v. 2.1)(BioBam BioInformatics, Valencia, Spain) annotation workflow. Briefly, BLASTp searches were carried out against the NCBI nr database (https://blast.ncbi.nlm.nih.gov/Blast.cgi, accessed on 9 August 2022) [97] using default parameters. We restricted searches to the Viridiplantae taxon. Simultaneously, searches for protein functional classification were carried out using InterProScan 5 [98,99]. We mapped the search results to the peptide sequences followed by the annotation using the default settings of the OmicsBox (v. 2.1). We performed GO term enrichment analyses on the significantly upregulated genes at both time points with Fisher’s exact test and false discovery rate (FDR)<0.05. OmicsBox (v.2.1) was used to perform annotations and GO statistics. 

The gene families were assigned using a locally executed uniport database. Amino acid sequences of the genes were extracted using TBTools (v.1_098722) and local blast database of UNIPROT (release-2022_01) was generated using BLAST+ executables (v.2.3.10+) [100]. A BLASTp search of DEGs was carried out against the UNIPROT database and, based on the resulting family names, were assigned using the UNIPROT ID mapping tool (https://www.uniprot.org/id-mapping, accessed on 9 August 2022). Protein kinases were identified using the integrated annotations for eukaryotic kinases, phosphatases, and phosphoprotein-binding domains (iEKPD) database online browsing tools (http://iekpd.biocuckoo.org/kinase_family_index.php, accessed on 9 July 2022) [77]. 

### 4.4. Isoform Analysis

We built a Kallisto (v.0.46.2) transcriptome index using the working model transcripts that include both high-confidence and working gene models of Castle Russet Genome Assembly (ver. 2.0) from SpudDB (http://spuddb.uga.edu/phased_tetraploid_potato_download.shtml, accessed on 9 August 2022). Transcript abundance and estimates were calculated also using Kallisto (v.0.46.2) [101]. Transcript isoform analysis was carried out using IsoformSwitchAnalyzeR (v.1.18.0) [102]. Transcript expression values were imported from Kallisto into IsoformSwitchAnalyzeR using the importRdata function [102]. The isoform switch test was carried out using DEXSeq implemented in IsoformSwitchAnalyzeR [102,103,104,105]. Then, predictions of premature termination codons (PTC) and thereby NMD-sensitivity were carried out [106,107]. The coding potentials of the transcripts were analyzed using CPC2 [108]. Domain architectures of the resulting proteins were identified using the Pfam database [109]. The presence of the signal peptides was inquired using SignalP (v.5.0) [110] and protein disorder was assessed using IUPred2A [87]. Predictions of the consequences of isoforms were conducted and visualized using IsoformSwitchAnalyzeR (v.1.18.0) [87].

### 4.5. Comparative Analysis of TM-Induced DEG in Potato and Arabidopsis

The DEGs following TM (5 ug/mL) treatment were compared (http://www.pnas.org/lookup/suppl/doi:10.1073/pnas.1419703112/-/DCSupplemental—last access date 10 April 2022) [25]. We performed Reciprocal Blast between potato and Arabidopsis DEGs to identify potential homologs genes with an > E-value of 1e^−5^ using NCBI blast v. 2.13.0+ and we retained only the best hits. An UpSet plot was generated using TBTools (v.1_098722). 

## 5. Conclusions

The data presented in this study show that TM treatment of leaves induces ER stress in a manner that causes the cell to tailor its transcriptome to respond to environmental challenges. This is evidenced by changes in gene expression, isoform usage, epigenetic regulation, RNA metabolism, translation, protein folding, and secretion. By comparing data in this study with previous data for Arabidopsis treated with TM to study transcriptomic responses, we identified common changes in gene expression but also many changes that are unique to potato plants. The pattern of gene induction is enough different that a broader set of time points may be needed to see the canonical changes affecting UPR expression as often reported for Arabidopsis. On the other hand, the early changes in gene regulation affecting downstream protein quality control include a wide range of genes that have not been reported before in Arabidopsis. Perhaps the slower induction of UPR in potato leaves makes it easier to capture transient information that may be occurring at a rapid pace in Arabidopsis that cannot be so easily captured in a transcriptomic study. These data open new models to describe ER stress responses in a solanaceous host. 

## Figures and Tables

**Figure 1 ijms-23-13795-f001:**
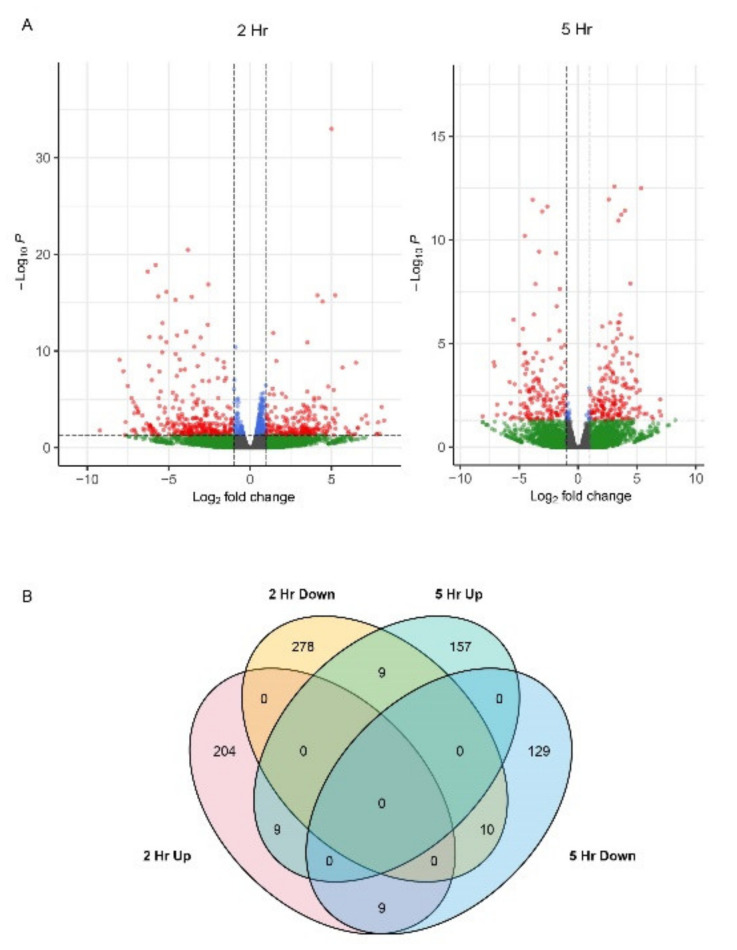
Numbers of differentially expressed genes (DEGs). (**A**) In the volcano plots, the red dots represent the DEGs with the log2FC threshold of ±1.0 (*p* < 0.05). Gray, green, and blue indicate genes that are not significantly altered in expression at Log 2FC 1.0 cutoff. (**B**) A Venn diagram highlighting unique and common DEGs in TM-treated leaves at 2 and 5 h.

**Figure 2 ijms-23-13795-f002:**
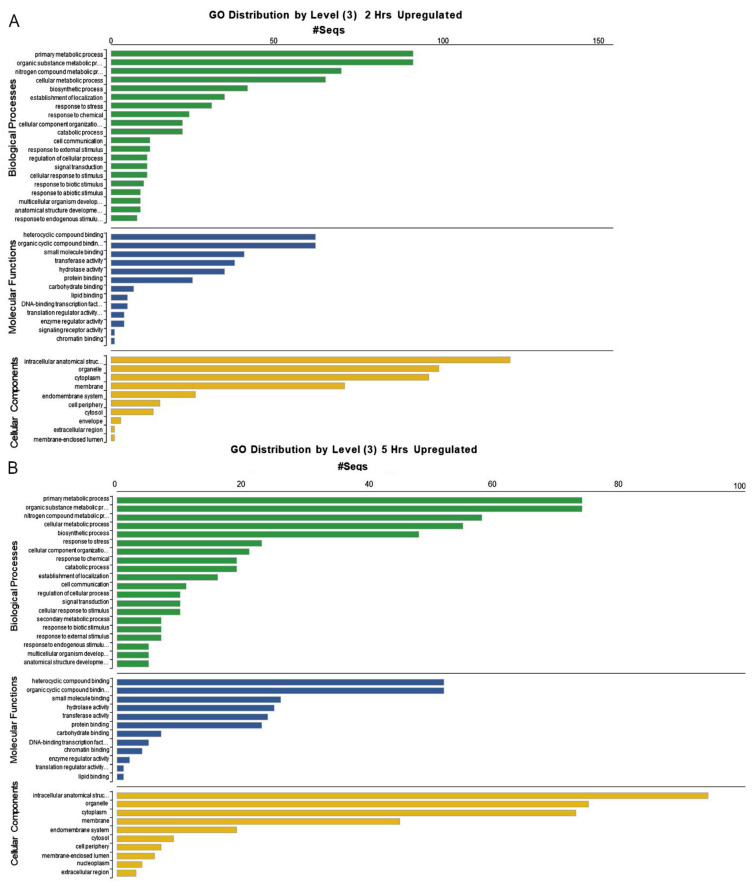
Level 3 Gene Ontology distribution of DEGs. Charts show the upregulated genes in TM-treated leaf tissues at (**A**) 2 and (**B**) 5 h.

**Figure 3 ijms-23-13795-f003:**
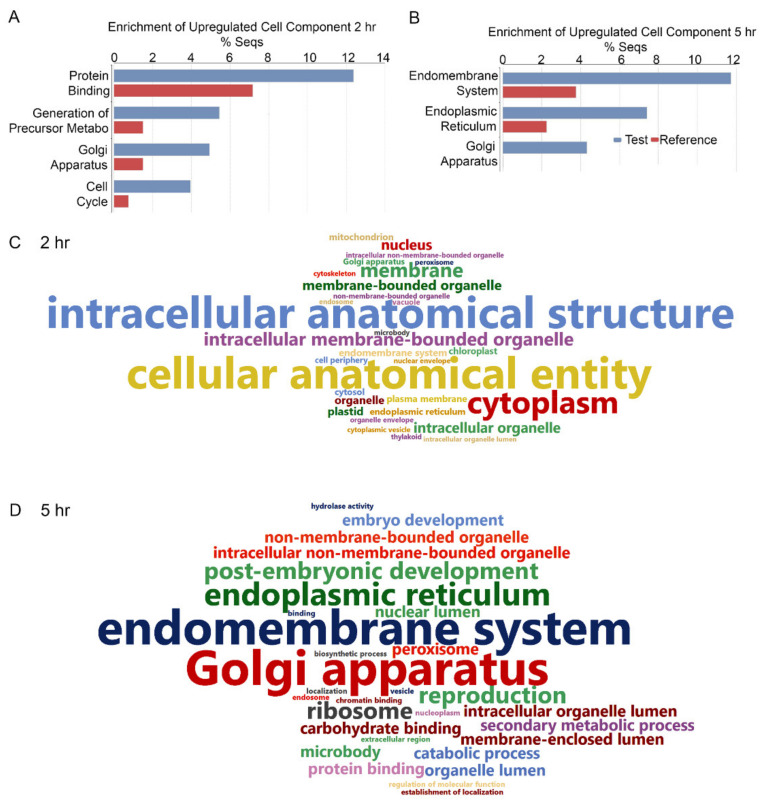
Functional enrichment of the cellular component of upregulated genes. Upregulated genes at (**A**) 2 and (**B**) 5 h; *p* < 0.05 using Fisher’s exact test. Word Clouds present detailed cellular components at (**C**) 2 and (**D**) 5 h after the TM treatment.

**Figure 4 ijms-23-13795-f004:**
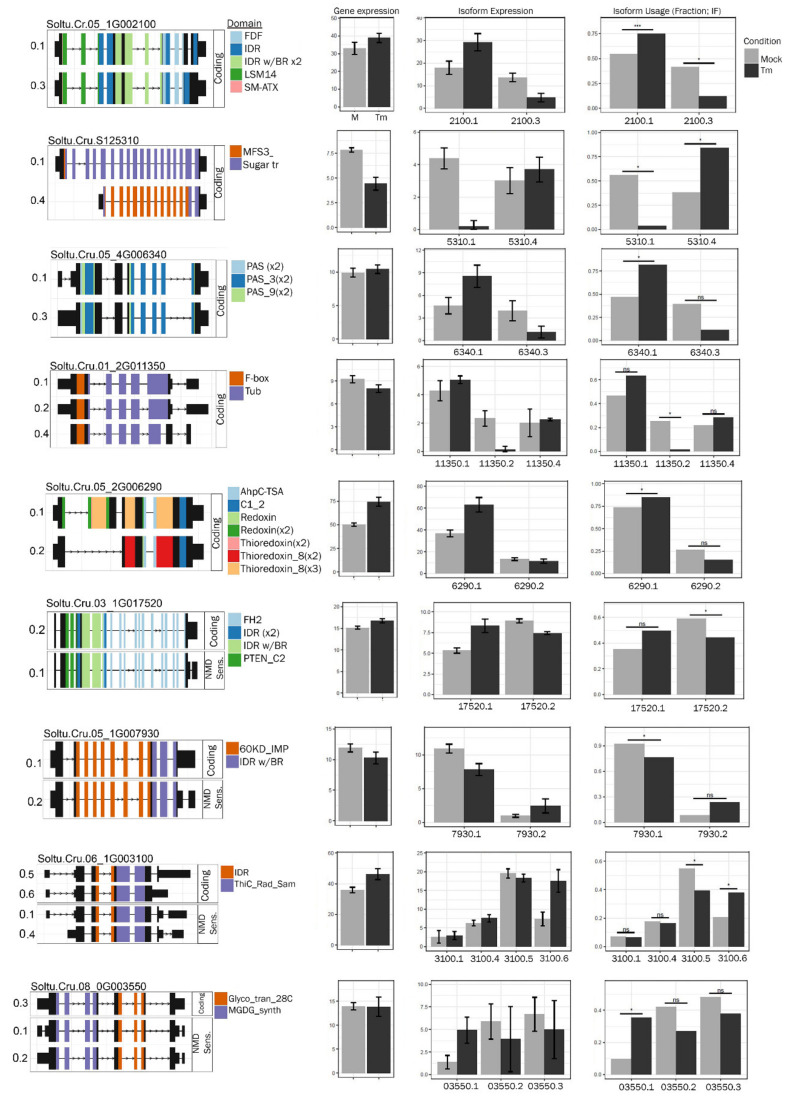
Isoform enrichment analysis at 2 h. Selected isoforms and their differential expression. The coding potential, NMD sensitivity, and functional PFAM domains are indicated. *—*p* < 0.05, ***—*p* < 0.001, ns—not significant.

**Figure 5 ijms-23-13795-f005:**
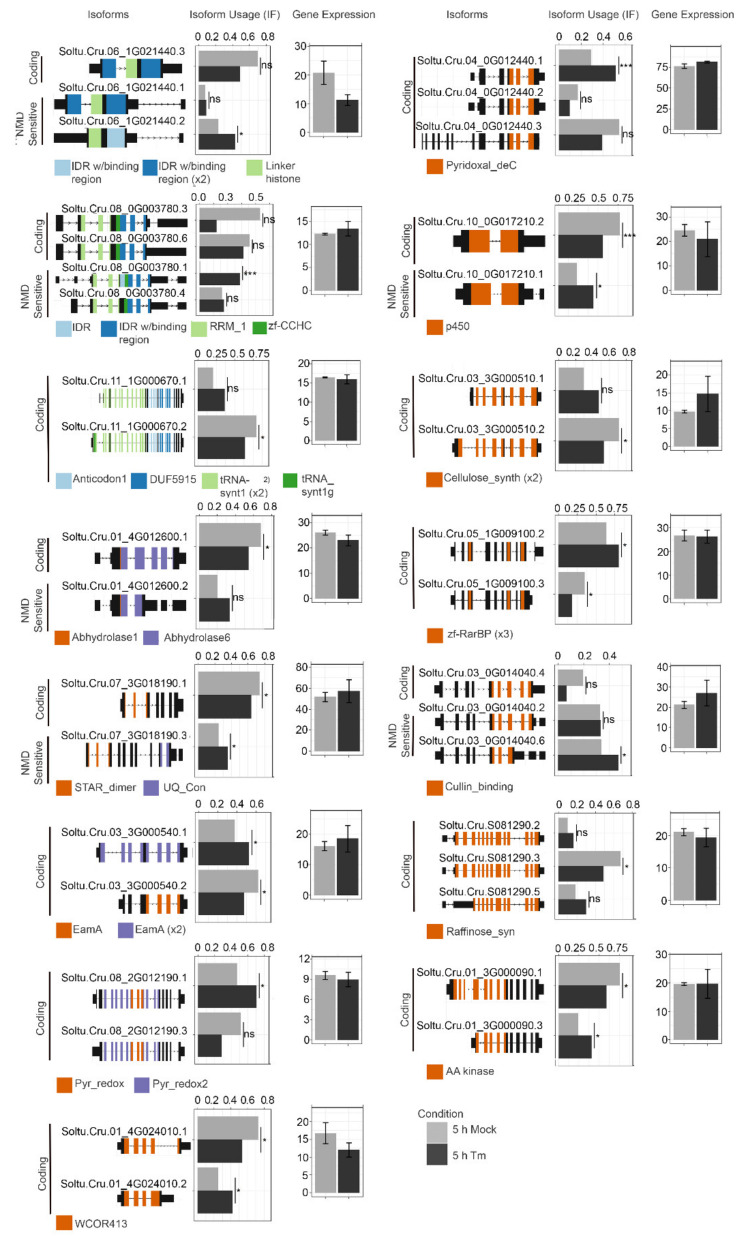
Isoform enrichment analysis at 5 h. Selected isoforms and their differential expression are shown. Coding potentials, NMD sensitivities, and functional PFAM domains are indicated. *—*p* < 0.05, ***—*p* < 0.001, ns—not significant.

**Figure 6 ijms-23-13795-f006:**
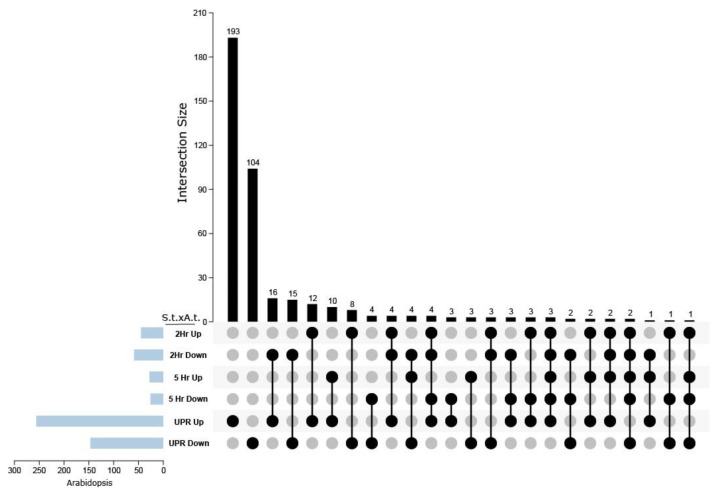
UpSet plot displaying the comparative analysis between UPR-related DEGs in Arabidopsis and potato leaves in response to TM treatment. The UpSet plot shows potato DEGs (2 and 5 h) that were also represented in a published Arabidopsis UPR DEGs dataset obtained following induction by TM. *S.t*. x *A.t.* is shorthand for *S. tuberosum* by *A. thaliana* comparison of DEGs.

## Data Availability

RNA-seq data were deposited in the Sequence Read Archive database at NCBI under the bio project PRJNA865435.

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
