# Peer review of "Comprehensive Transcriptome Analysis Reveals Genome-Wide Changes Associated with Endoplasmic Reticulum (ER) Stress in Potato (Solanum tuberosum L.)"

_ijms, 2022, doi:10.3390/ijms232213795_

Round 1
Reviewer 1 Report
The manuscript is well-written, I recommend it to accept.
Author Response
Thank you.
Reviewer 2 Report
Dear Authors,
I have reviewed your manuscript "Comprehensive transcriptome analysis reveals genome-wide changes associated with ER stress in Potato (Solanum tuberosum L.)", submitted for publication in International Journal of Molecular Sciences.
My main impression is that your manuscript is very well written and that your research is conducted properly, with appropriate conclusions. All of my recommendations for correction thus qualify as minor revisions, and are given as follows:
· English language – the language of the manuscript calls for thorough revision
o the use of singular and plural forms of verbs ("was/were", e.g., in lines 646, 656-661, etc.)
o the use of italic letters – genus names (Arabidopsis) should always be written in italics, whereas higher-rank taxa (such as Viridiplantae) should be written in non-italics; p as designation for confidence level (e.g., line 141) should be written in italic, etc.
o British and American English – since you kept American spelling throughout most of your manuscript, I would suggest spelling "orthologs" and "homologs" in American spelling instead of British ("orthologues, "homologues"), please see lines 37, 73, 139, 445
o throughout the manuscript, there are numerous lapsuses with words replaced by similarly sounding words, such as:
§ "shows the" instead of "shows THAT" (line 177)
§ "World clouds" instead of "word clouds" (line 193)
§ "coregulation" instead of "coregulator" (line 269)
§ "zing finger" instead of "zinc finger" (line 301) etc.
§ "grounded" means "prevented from flying" (from "to ground"), whereas you meant that the leaves were ground, from "to grind".
· Figures 2 and 3 contain blurred lettering that is not readable. Based on my experience as both author and reviewer this might be a fault of the MDPI-derived pdf rather than a fault of the Authors, however everyone should try to make sure that the lettering in the published version of the article is clear and readable. Also, the bottom part of Figure 3B seems to be cut off, please double-check.
· Portions of the text are inappropriately placed within the manuscript, and should be relocated for a more appropriate manuscript structure:
o lines 107-115 (starting from "Foliar TM sensitivity...") – the part of the text regarding the application of TM in Arabidopsis research should be moved to the earlier parts of the Introduction (before you mention potato for the first time). When you start mentioning potato, the rest of the Introduction should be dedicated solely to potato, so as to introduce the goal of your current research. Especially the closing paragraph of the Introduction is a place that should be narrowly dedicated only to presenting the goal of the current research.
o The Results section contains many parts citing previous research, including research performed by other research groups. While this is not always inappropriate (for instance when you need to clarify a result immediately without actually discussing it), it should be however limited to minimum usage, otherwise parts of your Results section will sound like an Introduction or Discussion of your (or even other people's) work. I recommend that you consider transferring the lines 346-356 and 362-368 to the Introduction, and that you look for similar portions of text that are within your Results section, and that you consider moving some, or all of them, into the Introduction or Discussion, as appropriate.
o line 43: Please start a new paragraph with "The inositol requiring enzyme (IRE1)..."
o line 51: Please start a new paragraph with "Alongside the IRE1/bZIP60 pathway in plants..."
· Other minor remarks:
o line 12: please add the word "protein" (PROTEIN folding capacity)
o lines 124-126: In these two sentences you mention the alignment rate twice, once as "being greater than 90%" and the other time as "being greater than 94%". Please revise to mention it only once.
o line 328: were the 28 factors upregulated, or differentially regulated at 2 hours and 5 hours? Please be specific.
o line 421-422: Please add a reference to the Supplementary Table, or write "data not shown", depending whether the results that you describe here are, or are not show in the Supplementary Table.
o line 480-481: It might be possible, though, that these genes actually are upregulated early, but their upregulation might be either terminated within the first 2 hours, or it might start later than 2 hours and end sooner than 5 hours after the TM treatment. If this possibility is reasonable, you should mention it in this sentence.
o line 581: the cultivar is called Russet Norkotah (with h in the end). Here you should add the word "plantlets" ("Russet Norkotah plantlets") and specify how old the plantlets were, i.e., in terms of micropropagation (such as, 20 days after micropropagation, or similar).
o line 583: Which three leaves? I suppose that you always sampled the three leaves according to a specific criterium (e.g., the three youngest leaves, or third, fourth and fifth youngest leaf, or similar).
o line 642: Please double-check this sentence, its structure seems strange, like a word or a punctuation sign might be missing from it.
o Author Contribution Statement: should not be written in quote marks.

Author Response
As requested we went through the manuscript carefully to improve verb tense, singular and plural forms throughout the manuscript.
Use of italics: we carefully went through and provide genus names, and gene names in italics as appropriate and eliminated inappropriate italics. The p for p value was italicized where used.
British/American English: We corrected spelling to be all American English. Orthologs, etc.
These specific changes were pointed out and addressed: § "shows the" instead of "shows THAT" (line 177). § "World clouds" instead of "word clouds" (line 193). § "coregulation" instead of "coregulator" (line 269). § "zing finger" instead of "zinc finger" (line 301) etc., § "grounded" means "prevented from flying" (from "to ground"), whereas you meant that the leaves were ground, from "to grind".
Text in figures: Reviewer 2 wanted text in Fig 2 and 3 improved for readability. We went through every figure to improve the text and improve readability throughout.
Moving portions of the text to other parts of the manuscript: This was a very astute observation and I appreciate the reviewer pointing this out. We transferred lines 346-356 and 362-368 out of the results to the introduction and then improved the writing so it fits better. Lines 107-115 was moved to earlier in the introduction. Both comments led us to reshape paragraphs of the introduction for better reading and these are highlighted. We changes lines 480-481; 581; 583; 642; and author contributions as requested.
Reviewer 3 Report
Dear authors, congratulations on the quality work. The paper is interesting, relevant, and well designed. In the present form paper is accepted
Author Response
Thank you
Reviewer 4 Report
The manuscript by Herath and Verchot reports on the gene expression profiling in potato to identify genome wide changes associated with the UPR, and identified processes like chromatin remodeling, transcriptional reprogramming, as well as changes in the structural components of the endomembrane network system as the important plant responses to TM. The data has been comprehensively and meaningfully analyzed. This study can thus act as a basis for further elucidating in detail the ER stress responses in potato.
I have a few concerns though:
1. The authors could consider functionally analyzing any differentially expressed gene identified in their study to show its involvement in UPR which would validate to some extent their findings.
2. Authors have identified chromatin modifications as an important response to ER stress. Authors should correlate the expression of chromatin remodeling factors with the expression of other differentially expressed genes identified in their study and discuss them well. In fact, it would be great if authors could show the chromatin modifications in a couple of highly differentially regulated genes (NOT in the chromatin remodeling factors themselves) that they think are highly potential candidates.
3. Have authors found a gene whose splice variants show differential regulation under TM and mock treatments?
4. There are several minor errors in writing. For example, Line 64: delete "()"; line 657: no space "CPC2" and "[". Pls check the whole manuscript for such mistakes and correct them.
5. Quality of figures needs to be greatly improved.
Author Response
Comment 1. The authors could consider functionally analyzing any differentially expressed gene identified in their study to show its involvement in UPR which would validate to some extent their findings.
Thank you for the suggestion. We have done the comparisons of the DEGs identified with already published set of Arabidopsis UPR genes. In future, we are planning to analyze the identified genes in order to further characterize their roles in UPR.
Comment 2. Authors have identified chromatin modifications as an important response to ER stress. Authors should correlate the expression of chromatin remodeling factors with the expression of other differentially expressed genes identified in their study and discuss them well.
We did our best to improve the discussion based on these comments and these are highlighted in yellow
Comment 3. Have authors found a gene whose splice variants show differential regulation under TM and mock treatments?
Interestingly, the genes coding for the identified splice variants were not significantly differentially expressed. This is pointing towards the differentially usage of isoforms under UPR stress without changing the overall gene expression.
- There are several minor errors in writing. For example, Line 64: delete "()"; line 657: no space "CPC2" and "[". Pls check the whole manuscript for such mistakes and correct them.
We did this.
- The quality of the figures needs to be greatly improved.
This is due to the MDPI pdf rendering. We have provided high-resolution images separately so MDPI can (hopefully) address the issue. However, we did introduce higher-resolution images into the text and we hope this is sufficient.
Reviewer 5 Report
Dear authors
This manuscript is regarding transcriptome analysis associated with ER stress in Potato. The authors have conducted the experiments with clear objectives, and the manuscript might be a good contribution to researchers working in this area. Authors have depicted differentially expressed the number of genes for UPR and their results corroborated with previously studied, for instance, tunicamycin treatment to Arabidopsis. Thus, I would like to recommend this manuscript for publication after minor revision.
A few of my comments are below for advancement of the manuscript:
As my suggestion, the Authors can perform the q PCR for a few selected genes for further confirmatory expression. Although, this study can be carried out later, not mandatory for this publication.
Figure 2 and 3: In both figures, the resolution of the image need to increase, since the image looks blurred.
Line no. 432: All scientific names should be in italic font. In the references section also. Such as references 21and 100 scientific names should be in italic.
Author Response
Comment 1: Figure 2 and 3, the resolution of the image need to increase, since the image looks blurred.
We have done our best to improve the images. Please see above response to reviewer
Comment 2: Line no. 432: All scientific names should be in italic font. In the references section also. Such as references 21and 100 scientific names should be in italic.
Fixed this.
Round 2
Reviewer 4 Report
none
Author Response
Thank you